# *Cannabis sativa* CBD Extract Shows Promising Antibacterial Activity against *Salmonella typhimurium* and *S. newington*

**DOI:** 10.3390/molecules27092669

**Published:** 2022-04-21

**Authors:** Logan Gildea, Joseph Atia Ayariga, Olufemi S. Ajayi, Junhuan Xu, Robert Villafane, Michelle Samuel-Foo

**Affiliations:** 1The Microbiology Program, College of Science, Technology, Engineering, and Mathematics (C-STEM), Alabama State University, Montgomery, AL 36104, USA; lgildea5810@myasu.alasu.edu (L.G.); rvillafane@alasu.edu (R.V.); 2The Biomedical Engineering Program, College of Science, Technology, Engineering, and Mathematics (C-STEM), Alabama State University, Montgomery, AL 36104, USA; 3Department of Biological Sciences, Alabama State University, Montgomery, AL 36104, USA; jxu@alasu.edu (J.X.); mfoo@alasu.edu (M.S.-F.)

**Keywords:** *Salmonella*, novel antibacterial agents, membrane disruption, cannabidiol

## Abstract

Products derived from *Cannabis sativa* L. have gained increased interest and popularity. As these products become common amongst the public, the health and potential therapeutic values associated with hemp have become a premier focus of research. While the psychoactive and medicinal properties of *Cannabis* products have been extensively highlighted in the literature, the antibacterial properties of cannabidiol (CBD) have not been explored in depth. This research serves to examine the antibacterial potential of CBD against *Salmonella newington* and *S*. *typhimurium*. In this study, we observed bacterial response to CBD exposure through biological assays, bacterial kinetics, and fluorescence microscopy. Additionally, comparative studies between CBD and ampicillin were conducted against *S*. *typhimurium* and *S*. *newington* to determine comparative efficacy. Furthermore, we observed potential resistance development of our *Salmonella* spp. against CBD treatment.

## 1. Introduction

*Cannabis sativa* L., a member of the genus *Cannabis* and a species in the family *Cannabaceae*, originated from Central Asia and is one of the oldest psychoactive plants known to humans [1]. Since its discovery, *C. sativa* has been utilized recreationally, medicinally, and industrially for numerous applications. Early uses of *C. sativa* focused on its use as an industrial material to produce textiles, ropes, and paper products [2,3,4]. Throughout early history, the use of *C. sativa* as an industrial product spread from Asia throughout Europe and Africa [5]. As industrial use of *C. sativa* became common, its use as a therapeutic against rheumatic pain, intestinal constipation, disorders of the female reproductive system, malaria, and other common health problems can be traced back to ancient China where it was one of the oldest pharmaceuticals recorded [6]. Over time, the use of *C. sativa* as a therapeutic option was introduced across the western world and by the 19th century had gained attention from medical science. The first clinical conference on *C. sativa* in 1860 led to an increased interest in the research and development of medicinal products derived from *C. sativa* [2]. This interest in *C. sativa* drastically declined in 1942 when it was removed from the United States Pharmacopoeia and lost its status due to research suggesting a correlation between *C. sativa* and ‘insanity’ [7]. This decision eventually led to the United Nations designating *Cannabis* as an illegal psychotropic substance in 1971 at the Convention on Psychotropic Substances [8,9]. These designations and findings drastically reduced research progression on *Cannabis* and produced a negative stigma on the substance from the perspective of the public [10,11].

As we move towards the mid-21st century, the acceptance of *Cannabis* and the potential of its byproducts has increased [12]. Continued research demonstrated the efficiency of *Cannabis* in the treatment of medical conditions including epilepsy, multiple sclerosis, Tourette’s syndrome, and other neurological diseases [13,14,15]. Additionally, research has further characterized the active compounds—tetrahydrocannabinol (THC) and cannabidiol (CBD)—of *Cannabis* and their psychological and physiological effects on humans [16,17,18,19]. While clinical research has primarily focused on the efficacy of *Cannabis* against neurological disorders, a current gap in knowledge is the efficacy of *Cannabis* and its byproducts as antibacterial agents. The current literature does suggest that *Cannabis* and more specifically its active compound, CBD, shows antibacterial function [20,21]. However, further research and characterization of this antibacterial function is crucial to the development of novel therapeutics against clinically relevant bacteria [22,23].

The increased prevalence and occurrence of antimicrobial resistance is a major health concern worldwide, making the development of alternative therapies a necessity [24,25,26,27]. The steady decrease in antibiotic efficacy alongside the decreased development of new antibiotics present a major obstacle in the treatment of multidrug-resistant bacteria [28,29]. Antibiotic-resistant and multidrug-resistant bacterial infections account for roughly 2.8 million infections and 35,000 deaths annually in the United States alone [30]. Antibiotic-resistant bacterial infections, while a major threat to public health, also present a major economic burden to the United States, resulting in an annual cost of USD 7.7 billion [31]. As resistant bacterial species persist, the development of novel antibacterial treatment methods is essential to the preservation of public health. The purpose of this research is to determine if CBD extracted from *C. sativa* demonstrates antibacterial activity against a common Gram-negative bacterial pathogen, *Salmonella*. CBD, initially not recognized as an active compound of cannabinoids, has been shown to possess potential therapeutic benefits [32,33,34]. *Cannabis* is commonly associated with psychotic effects. We now know that these effects can be attributed to THC, whereas CBD “exhibits no effects indicative of any abuse or dependence-potential and to date, there is no evidence of public health related problems associated with the use of pure CBD” according to the World Health Organization [35]. The literature suggests that CBD’s antibacterial function is carried out through the disruption of the cell membrane of both Gram-positive and Gram-negative bacteria [36]. The antibacterial activity of CBD against several bacterial pathogens including *Staphylococcus aureus*, *Streptococcus pneumoniae*, *Clostridiosis difficile*, *Neisseria* spp., *Moraxella catarrhalis*, and *Legionella pneumophila* has been observed [20,21]. Study of CBD as an antibacterial agent requires further determination of the range of bacteria that CBD exhibits antibacterial activity against. Examining the potential of CBD against a clinically relevant pathogen such as *Salmonella* is essential to further expand our knowledge on CBD as a potential novel therapeutic.

*Salmonella* species are one of the most common and prevalent foodborne pathogens worldwide, found in several food products including poultry, seafood, and other fresh or processed meats [26,27,37,38,39]. *Salmonella* accounts for 1.35 million infections, 26,500 hospitalizations, and 420 deaths annually in the United States alone [30]. *Salmonella* is a significant spoilage hazard. In addition, the increased prevalence of multidrug-resistant strains makes *Salmonella* a major threat to public health [25,30]. The CDC reported in 2019 that *Salmonella typhimurium*, one of the species examined in this study, accounted for 59% of ampicillin-resistant *Salmonella* infections in the United States [30]. The continued use of broad-spectrum antibiotics could drastically increase the prevalence of antibiotic and multidrug-resistant *Salmonella* strains, making the development of novel antibiotic alternatives a necessity [31,40].

In this study, we performed plate assays, fluorescence microscopy, and growth kinetic studies to determine the antibacterial activity of CBD extracted from *C. sativa* against *S*. *typhimurium* and *S*. *newington*. Additionally, we conducted comparative kinetic studies of the two *Salmonella* strains in the presence of CBD or a common broad-spectrum antibiotic, ampicillin. Finally, we examined resistance development of *S. typhimurium* and *S. newington* against CBD treatment over an extended time. We hypothesized that CBD would exhibit antibacterial activity against *S*. *typhimurium* and *S*. *newington*. The results of these studies suggest that CBD does exhibit antibacterial activity against these *Salmonella* species, further encouraging research and development of CBD as a potential antibacterial agent.

## 2. Results

### 2.1. Gas Chromatography

For confirmation of pure CBD in our *C. sativa* extract, gas chromatography (GC) was conducted. Figure 1A represents the internal standard for CBD and THC, while Figure 1B represents our sample. In these readings, we observe a strong single peak at the 8.23 min retention time (RT) in our sample. This is consistent with the expected RT of CBD at 8.22 min as shown in the internal control sample. The absence of other peaks represents the absence of other compounds or contaminants. Overall, we conclude through the GC analysis that our sample is pure CBD.

### 2.2. Plate Assays

To examine the potential antibacterial activity of our CBD against *S. typhimurium* and *S. newington*, the Kirby–Bauer and spot assays were conducted. These assays allow us to visualize and quantify the inhibitory effects of CBD against our *Salmonella* strains. Both assays confirmed inhibitory activity of CBD against *S. typhimurium* and *S. newington*. In the Kirby–Bauer assay, we observed zones of inhibition (ZOI) around the CBD-treated disks suggesting inhibition of bacterial growth due to exposure to CBD (Figure 2C). These results also suggest a dose-dependent inhibition due to CBD treatment with ZOIs decreasing as CBD concentration decreased (Figure 2A,B). The results of the spot assay further confirmed the dose-dependent nature of CBD’s inhibitory activity. As CBD concentration decreased, we saw an increase in the density of bacterial colonies in both *S. typhimurium* and *S. newington* (Figure 2D). These data also suggested that *S. newington* was more susceptible to CBD treatment than *S. typhimurium* due to the larger ZOIs (Figure 2B) and lower colony density (Figure 2D).

### 2.3. CBD Extract Reduces Bacterial Growth of Salmonella typhimurium and Salmonella newington

To determine the effect of CBD on *S. typhimurium* and *S. newington*, cultures were treated with CBD at concentrations of 1.25, 0.125, 0.0125, or 0.00125 μg/mL. The optical density (OD600) was recorded using a spectrometer (Molecular Devices SpectraMax^®^ ABS Plus) hourly for 6 h. Both *S. typhimurium* and *S. newington* cultures treated with CBD showed a significant reduction in OD600 within 6 h of treatment (Figure 3A,B). It was even observed that in *S. newington*, CBD at a concentration as low as 0.0125 μg/mL reduced bacterial growth over the 6 h period. These results suggest that *S. newington* might have a greater susceptibility to CBD than *S. typhimurium*, whose MIC was 0.125 μg/mL. It was observed that treatment of *S*. *typhimurium* with CBD at a concentration of 0.00125 μg/mL resulted in a higher OD600 than that of the control treatment. We infer that these results suggest a rapid development of resistance to low concentrations of CBD treatment resulting in an increase in bacterial fitness. The reduction of OD600 signifies that CBD extract does exhibit antibacterial characteristics. The antibacterial effect on *S. typhimurium* and *S. newington* was observed to be dose-dependent with the OD600 increasing in correlation with decreased concentrations of CBD.

### 2.4. Assessment of Bacterial Membrane Integrity via Fluorescent Staining of CBD-Treated Salmonella Cells

To examine the effect of CBD treatment on *Salmonella* cells, *S. typhimurium* and *S. newington* samples were treated with CBD at concentrations of 1.25, 0.125, or 0.0125 μg/mL. These samples were visualized using immunofluorescent 4′,6-diamidino-2-phenylindole (DAPI) staining. This staining technique emits fluorescence when the stain binds to AT-rich DNA. This stain is membrane impermeable, and thus, fluorescence represents compromised membrane integrity. DAPI staining confirmed that CBD treatment resulted in degradation of membrane integrity after 5 min and 30 min of treatment (Figure 4A,B). Quantitative analysis using densitometry of these images showed increased DAPI fluorescence from the 5 min to the 30 min time points (Figure 4C). These images all indicate that CBD can successfully damage *Salmonella* cells outer lipopolysaccharide membranes.

### 2.5. Comparative Study of CBD and Antibiotic Treatment against Salmonella

*Salmonella* infections are typically treated with broad-spectrum antibiotics such as ampicillin; however, the development of resistance to these treatments has become more prevalent, thus increasing the need for alternative treatments. To compare conventional antibiotics to CBD, we conducted comparative kinetic studies to observe antibacterial activity. In this study, *Salmonella* strains *S. typhimurium* and *S. newington* at an OD600 of 0.5 were treated with the minimum inhibitory concentration (MIC) of ampicillin (0.5 µg/mL) or CBD (0.125 µg/mL). Results suggested that CBD and ampicillin both successfully inhibited *Salmonella* growth in both strains (Figure 5). In comparison, both treatments resulted in a similar OD600 after 6 h of treatment suggesting that CBD was able to inhibit bacterial growth to an extent similar to ampicillin.

### 2.6. Developed Resistance of Salmonella to CBD Treatment

A major concern with conventional antibiotic treatments is the development of resistance. To observe the development of resistance in *Salmonella* against CBD treatment, extended kinetic studies were conducted over a span of 48 h. *Salmonella* strains *S. typhimurium* and *S. newington* were treated with ampicillin (MIC 0.5 µg/mL) or CBD (1.25, 0.125, 0.0125, or 0.00125 μg/mL). Results confirmed a developed resistance to CBD in both *Salmonella* species (Figure 6). At concentrations of 1.25 and 0.125 μg/mL of CBD, resistance development was not observed until the 48 h time point. These results suggest that CBD was effective in killing *Salmonella* and decreasing the rate of resistance development in both *Salmonella* strains over the initial 24 h time period. At 0.0125, 0.00125, and 0.000125 μg/mL concentrations of CBD, there was no significant reduction in bacterial growth and resistance was developed after only 12 h of exposure in both *Salmonella* strains. These results suggest that both *Salmonella* strains were able to develop resistance quickly to low concentrations of CBD.

### 2.7. CBD Effectiveness against Salmonella Biofilm

The literature has determined that most hospital-acquired infections are the result of bacterial biofilm formation [41]. Typically, a result of bacteria reaching a non-replicatory state, biofilms are a major health threat that are characterized by antibiotic resistance, reoccurring infection, and sepsis [42]. Understanding CBD activity against bacterial biofilm is an important aspect of determining CBD viability against *Salmonella* spp. To determine the efficacy of CBD treatment against biofilm formation, *S*. *typhimurium* biofilms were grown in vitro and treated with CBD at a concentration of 0.125 µg/mL. *S. typhimurium* biofilms were then visualized through scanning electron microscopy (SEM) (Figure 7).

## 3. Discussion

The rapid increase of antibiotic- and multidrug-resistant bacteria over the early 21st century is a major threat to public health [30]. Knowing this, the study of potentially efficacious alternative therapeutics has been a topic of growing interest. *C. sativa* products, while typically associated with the treatment of neurological disorders, have shown promise as antibacterial agents against several notable pathogens [20,21]. Of the multiple metabolites of *C. sativa*, research suggests that CBD is the most promising [32,33,34]. This compound, unlike THC, holds no psychoactive properties but does possess antioxidant, anti-inflammatory, and antibacterial properties [22,43,44]. CBD’s antibacterial potential has shown some promise; however, research is relatively limited. Further investigation and confirmation of the specificity of this antibacterial activity is crucial as the field of medicine searches for new viable therapeutics for resistant bacterial infections.

*C. sativa* products have exhibited a wide variety of applications in numerous fields. This study helps expand our knowledge of *C. sativa*-derived CBD as an antibacterial agent against two common Gram-negative pathogenic *Salmonella* strains. *Salmonella* spp. have significance in terms of resistance to antibiotics and standard treatment protocols making these infections a major threat to the preservation of public health [25,28,30]. Discovery and development of novel antibacterial agents such as CBD are a major step towards the future of therapeutics in a world where antibiotics are no longer efficacious and cost effective [20,21]. This study serves primarily to determine the antibacterial potential of CBD against *S. typhimurium* and *S. newington* through plate assays, fluorescence microscopy, and kinetic assays. Additionally, this study examines the comparative antibacterial activity of CBD and ampicillin as well as resistance development of *S. typhimurium* and *S. newington* against CBD treatments.

In this study, we confirm that CBD does exhibit antibacterial activity against *S. typhimurium* and *S. newington*. This inference was derived from a combination of plate assays, fluorescence microscopy, and bacterial growth kinetic assays and thus allowed us to propose a potential mechanism of this antibacterial activity. Plate assays consisted of the Kirby–Bauer assay and the spot assay, both of which served as initial confirmation of bacterial inhibition of *S. typhimurium* and *S. newington*. Once this inhibitory activity was observed, a fluorescence microscopy assay for membrane integrity was utilized. This assay utilized DAPI stain, a membrane-impermeable DNA binding stain. Fluorescence of the DAPI stain confirmed that CBD treatment of *S. typhimurium* and *S. newington* resulted in a loss of membrane integrity and cell death, and thus proving our hypothesis that the CBD treatment leads to membrane disruption. Other researchers have reported this mechanism as the antibacterial mechanism of action of CBD [36]. Finally, kinetic assays were conducted to examine the effect of CBD treatment on lag-phase *S. typhimurium* and *S. newington* over a span of 6 h. Once again, it was observed that CBD was effective in growth inhibition of both species in a 6 h period with a MIC of 0.0125 μg/mL in *S. typhimurium* and 0.125 μg/mL in *S. newington*. These findings confirm that CBD does possess antibacterial activity through mechanisms similarly described in other Gram-negative bacterial species [20,21,36].

Within this study, we were able to exhibit the antibacterial effectiveness of CBD at a concentration of 0.125 μg/mL against a *S. typhimurium* biofilm (Figure 7). This study is relevant in terms of clinical application of CBD where most hospital-acquired infections are in the form of biofilms [41]. Biofilms typically associated with antibiotic resistance result in infections that are difficult to treat with an increased chance of sepsis and mortality in some cases [42]. Considering the threat that biofilms pose, it is essential that novel therapeutics are developed for the prevention and treatment of these harmful infections. Our results suggest that CBD features antibacterial activity that is not only effective against lag- and log-phase *Salmonella* spp. but also against *Salmonella* biofilms.

An important facet of this study was the comparison between CBD and broad-spectrum antibiotic treatment. In this study, we examined the comparative kinetics between CBD treatment and ampicillin treatment of *S. typhimurium* and *S. newington*. We concluded from the results of these kinetic studies that both CBD and ampicillin at their MIC concentrations are successful at inhibition of *S. typhimurium* and *S. newington* growth. It is important to note that the MIC concentration of ampicillin (0.5 μg/mL) is significantly higher than the MIC of CBD (0.125 μg/mL) (Figure 5). To further investigate the comparative efficacy of a novel therapeutic agent (i.e., CBD in this study) and a standard treatment (such as ampicillin), it was important to examine the resistance development of the pathogen to the therapeutic agents. To examine the resistance of *S. typhimurium* and *S. newington* against CBD and ampicillin treatments, extended growth kinetics under different dosages of CBD were carried out over a 48 h period (see Figure 6). These results varied across the two strains of *Salmonella* that were used. In *S. typhimurium*, CBD concentrations of 1.25 and 0.125 μg/mL were able to inhibit bacterial growth over the span of 24 h after a single treatment. These results are similar to the growth kinetics we observed with ampicillin treatment after 24 h, whereas lower concentrations of CBD showed a rapid rise in growth following 12 h of treatment, suggesting a rapid development of resistance. After 48 h of exposure to CBD treatment, *S. typhimurium* had developed resistance, suggested by the rapid rise in OD600 at the 48 h time point, to all CBD treatments and the ampicillin treatment. This result is consistent with the literature on *S. typhimurium*, which accounts for 59% of all ampicillin-resistant *Salmonella* infections in the United States [30]. These results raise the question of what mechanism *S. typhimurium* develops to resist the antibacterial activity of CBD. The literature has suggested several mechanisms for *S. typhimurium* resistance to antibiotics including the multidrug efflux pump AcrAB, OXA-1 β -lactamase, and other beta-lactamase genes (*bla*). Ampicillin, a beta-lactam class antibiotic, kills bacteria through binding to penicillin-binding proteins in the cytoplasmic membrane [45,46,47]. This mechanism of resistance relies on the inner mechanisms of the bacterial cell, whereas we hypothesize that CBD’s antibacterial activity is a result of membrane disruption. In this work, we demonstrated that the antibacterial activity of CBD was generated through membrane integrity disruption. Therefore, we suggest that *S. typhimurium* resistance to CBD might be conferred through a resistance mechanism different from that of the ampicillin-resistance mechanism of this bacteria. This could be significant in the application of CBD as a therapeutic agent against ampicillin-resistant *S. typhimurium* or in a potential co-therapy designed with CBD and ampicillin. Co-therapies typically reduce the ability of bacteria to persist due to bacteria having to rapidly develop multiple resistance mechanisms [27,48,49,50,51]. Additionally, co-therapy strategies have been shown to ‘revert’ resistant bacteria back to a state that is susceptible to antibiotic treatment [52]. These factors further buttress the need to employ CBD in a potential CBD–antibiotic co-therapy once the resistance mechanism of CBD is fully understood. The extended kinetics of *S. newington* revealed that this strain was more susceptible to CBD treatments of 1.25, 0.125, and 0.0125 μg/mL 24 h after treatment. However, at the 48 h time point, there was a strong development of resistance in all CBD treatments but no resistance development to ampicillin. These results suggest that the mechanism of resistance is different between ampicillin and CBD (Figure 6). Further studies to characterize the resistance developed by *S. typhimurium* and *S. newington* against CBD are important to understand how to responsibly develop this potential therapeutic agent.

The previous literature has outlined the potential of CBD against several, mostly Gram-positive, bacterial pathogens including *Staphylococcus aureus*, *Streptococcus pneumoniae*, and *Clostridioides difficile* [20,21]. In these studies, there was a reported MIC of 1–5 μg/mL against *Staphylococci* spp., *Clostridioides* spp., and *Streptococci* spp. [20]. More recent studies have reported a MIC of 0.5–2 μg/mL against several clinical isolates of methicillin-resistant *Staphylococcus aureus* (MRSA). This study focused on a relevant Gram-negative pathogen, *Salmonella,* whose tolerance of CBD has not previously been studied. Within our study, *S. typhimurium* and *S. newington* were shown to have a MIC of 0.125–1.25 μg/mL. These MIC results are consistent with the previous literature exploring the antibacterial capabilities of CBD [20,21]. Results from this study and the studies mentioned earlier [20,21] seem to suggest bacterial susceptibility to CBD within a MIC range of 0.1–5 μg/mL. As we continue to explore potential antibacterial agents, it is essential that we explore the effectiveness of these agents against a multitude of diverse bacterial pathogens. Considering this, our study confirms that CBD does have antibacterial activity against two Gram-negative *Salmonella* strains, filling a valuable gap in our knowledge of CBD as an antibacterial agent. While this study illuminates the potential of CBD as a therapeutic and fills a void in the current literature, future work is necessary for further development of this bioactive compound as a therapeutic agent.

While our study was successful in determining the presence of CBD’s antibacterial activity against *S. typhimurium* and *S. newington*, there were limitations. One limitation was that the specific resistance mechanism of *S. typhimurium* and *S. newington* against CBD was not investigated in this study. This study simply determined the ability of our *Salmonella* strains to develop resistance against the CBD treatments administered and did not define the mechanism of resistance. The development of CBD as a novel therapeutic option will require further studies and characterization. Some relevant studies for the progression of CBD as a potential therapeutic include resistance mechanisms to CBD, cytotoxicity of CBD especially to co-therapy situations, immunological response to CBD treatment, CBD function in pathophysiological conditions, and in vivo models. These future studies will serve to further expand our knowledge of CBD as an antibacterial agent with potential therapeutic benefits.

## 4. Materials and Methods

### 4.1. CBD Extraction

CBD extraction was carried out by Sustainable CBD LLC. *C. sativa* biomass was weighed, tagged, and recorded in a receiving trailer for processing. Following storage, the biomass was reduced to between 200 and 500 microns and underwent CO_2_ extraction in an Apeks Transformer for subcritical extraction (Gibraltar Industries Inc., Buffalo, NY, USA). Subcritical extraction was carried out at a target pressure of 1200 psi, chiller temperature of 20–25 °C, propylene glycol percentage of 10%, orifice size of 22, resultant separator pressure of 350–400 psi, resultant separator temperature of −6–4 °C, for an extraction time of approximately 2–3 h. Following subcritical extraction, samples were prepared for decarboxylation prior to supercritical extraction. Hemp biomass was placed in the oven for approximately 100 min at 265 °C to decarboxylate. Once decarboxylation was carried out, an Apeks Transformer was utilized for supercritical extraction (Gibraltar Industries Inc., Buffalo, NY, USA). Supercritical extraction was carried out at a target pressure of 1800 psi, chiller temperature of 37–42 °C, propylene glycol percentage of 10%, orifice size of 18, resultant separator pressure of 350–400 psi, resultant separator temperature of 0–10 °C, for an extraction time of approximately 1–2 h per pound of material. The resulting material then underwent winterization through addition of ethanol to the crude extract. This sample was frozen and then filtered through Buchner funnels and the remaining ethanol was evaporated using a Heidolph rotary evaporator (Heidolph Instruments GmbH & Co.KG, Kelheim, Germany). Distillation was carried out using the Lab Society 5 L short path distillation unit (Lab Society^®^, Boulder, CO, USA) for further refinement. The resulting product of this procedure was winterized cannabinoid [53,54,55].

### 4.2. Gas Chromatography Analysis of CBD

CBD extracts were analyzed by gas chromatograph (GC)/mass spectrometry (MS) using Agilent 6890N GC and Agilent 5975 MS (Agilent Technologies Inc., Santa Clara, CA, USA) with a Restek Rxi-5Sil MS with integra guard column (15 m, 0.250 mmID, 0.25 µm df) (Restek Corporation, Bellefonte, PA, USA). A solution of Restek Qualitative Retention Time Index Standard (Restek Corporation, Bellefonte, PA, USA) was used to create the retention time index. The temperature of the injection port was set at 280 °C and the helium gas flow was constant at 1.1 mL/min. The samples were injected in split mode (2:1) with a volume of 1 µL of sample. The GC oven temperature was programmed as follows: initial temperature of 70 °C for 4 min, ramp to 200 °C at 20 °C/min, ramp to 300 °C at 8 °C/min, ramp to 325 °C at 50 °C/min with a 5 min hold, thus requiring a total run time of 30.5 min. The MS transfer line was set to 250 °C, source to 230 °C, and quads to 150 °C. The raw data were processed and analyzed using Agilent Enhanced ChemStation software (Agilent Technologies Inc., Santa Clara, CA, USA). The NIST 2.0 library was used with AMDIS for compound identification.

### 4.3. Plate Assays for Antibacterial Screening of CBD

To qualitatively determine the antibacterial potential of CBD against *S*. *typhimurium LT2* strain MS1868 (a kind gift from Dr. Anthony R. Poteete, University of Massachusetts) and *S*. *newington* (also known as *S*. *enterica serovar* Anatum *var 15+* strain UC1698, a kind gift from Dr. Sherwood R. Casjens, University of Utah), plate assays were utilized. Prior to the assays, CBD was serially diluted to the concentrations of 1.25, 0.125, 0.0125, or 0.00125 μg/mL. Bacterial strains *S. typhimurium* and *S. newington* were incubated until the late log-phase (OD > 1) in sterile Luria broth (BD Difco, Franklin Lakes, NJ, USA).

To conduct the Kirby–Bauer assay, plates were inoculated with overnight cultures of either *S. typhimurium* or *S. newington* and top agar (BD Difco, Franklin Lakes, NJ, USA) to create a bacterial lawn. These plates were then divided into quarters representing the four CBD dilutions (1.25, 0.125, 0.0125, or 0.00125 μg/mL). Sterile paper discs were soaked in a designated dilution of CBD or treated with dH_2_O (control) and then applied to the agar plates in triplicate. The results were three discs per CBD dilution on the agar plate. Plates were then incubated at 37 °C for 24 h, and bacterial growth was observed to determine zones of inhibition (ZOI) around the CBD-treated discs [56,57,58]. ZOIs were quantitatively assessed using ImageJ (NIH Image, Bethesda, Maryland). This assay was completed in triplicate.

For further confirmation of antibacterial activity, spot assays were conducted using lag-phase *S. typhimurium* and *S. newington* cultures (OD > 1). Sterile agar plates were inoculated with four 10 μL aliquots of *S. typhimurium* or *S. newington*. Each ‘dot’ was then inoculated with 10 μL of one of the four dilutions (1.25, 0.125, 0.0125, or 0.00125 μg/mL) of CBD. Each culture was also spotted and treated with 10 μL of dH_2_O as the control. The plate was then incubated for 24 h at 37 °C and observed for inhibition of bacterial growth [59]. This assay was completed in triplicate.

### 4.4. Fluorescence Microscopy of CBD-Treated Salmonella Cells

To visualize the effects of CBD treatment on *Salmonella* cells, fluorescence microscopy was utilized. *Salmonella* cells were fixed and stained using DAPI (4′,6-diamidino-2-phenylindole) to determine cell membrane integrity. DAPI stain binds exclusively to dsDNA, which is only accessible as a result of compromised membrane integrity and expulsion of the cytoplasmic material including the nucleus out of the cell. Visualization of DAPI fluorescence under the fluorescence microscope (Biotek Cytation™ 3 Automated Fluorescence Microscope) (Agilent Technologies Inc., Santa Clara, CA, USA) confirmed cell membrane damage and hence cell death. *Salmonella* cells were treated with CBD dilutions (1.25, 0.125, or 0.0125 μg/mL) or left untreated (control). Following treatment, cells were fixed at either 5 min or 30 min and assessed using fluorescence microscopy [27].

### 4.5. Bacterial Growth Kinetics

To study bacterial growth kinetics, a 96-well plate (Fisherbrand™, Fisher Scientific, Fair Lawn, NJ, USA) was inoculated with 180μL of overnight bacterial cultures of either *S. typhimurium* or *S. newington* at an OD600 ≈ 0.5. The cultures were then incubated at 37 °C with rotary shaking at 121 rpm. Measurements of bacterial density (OD600) were taken every hour for 6 h and again at the 12, 24, and 48 h time points using a spectrometer (Molecular Devices SpectraMax^®^ ABS Plus) (Molecular Devices LLC., San Jose, CA, USA). This experiment was completed in triplicate [27,60].

### 4.6. Bacterial Growth Kinetics in the Prescence of CBD

To study how CBD affects bacterial growth kinetics, three 96-well plates were inoculated with overnight bacterial cultures of either *S. typhimurium* or *S. newington* at an OD600 ≈ 0.5. These cultures were then treated with varying concentrations of CBD (1.25, 0.125, 0.0125, or 0.00125 μg/mL). The cultures were then incubated at 37 °C with rotary shaking at 121 rpm. Measurements of bacterial density (OD600) were taken hourly for 6 h and at the 12, 24, and 48 h time points using a spectrometer (Molecular Devices SpectraMax^®^ ABS Plus). This experiment was completed in triplicate [27,60].

### 4.7. Comparative Bacterial Growth Kinetics in the Prescence of Ampicillin or CBD

To study how ampicillin affects bacterial growth kinetics, three 96-well plates were inoculated with overnight bacterial cultures of either *S. typhimurium* or *S. newington* at an OD600 ≈ 0.5. These cultures were then treated with either ampicillin (0.5 μg/mL) or CBD (0.125 μg/mL). The cultures were then incubated at 37 °C with rotary shaking at 121 rpm. Measurements of bacterial density (OD600) were taken hourly for 6 h and at 24 and 48 h time points using a spectrometer (Molecular Devices SpectraMax^®^ ABS Plus). This experiment was completed in triplicate [27,60].

### 4.8. SEM Imaging of CBD-Treated S. typhimurium Biofilm

*S. typhimurium* was grown in 6-well plates (Fisherbrand™, Fisher Scientific, Fair Lawn, NJ, USA) supplemented with LB broth until a biofilm was developed. *S. typhimurium* biofilm samples were left untreated or treated with CBD at a concentration of 0.125 μg/mL, and the morphology and surface microstructure of the bacteria were observed using scanning electron microscopy. Untreated *S. typhimurium* served as control. Prior to microscopy, the specimens were dried in a vacuum, and sprayed with gold using an EMS Quorum (EMS 150R ES) ion-sputtering instrument and observed through an Analytical Scanning Electron Microscope (SEM) (JEOL JSM-6010LA, Tokyo, Japan) installed with IntouchScope software (JSM-IT200 InTouchScope™ SEM Series, JEOL Solutions, Tokyo, Japan). Sample preparation for SEM was carried out as described by Li et al., 2021 [41] with few modifications. In short, the specimens for SEM were fixed with 10% formaldehyde solution at room temperature for 10 min, washed with PBS solution thrice, and dehydrated serially in 50%, 70%, and 95% absolute ethanol solutions for 10 min each. Finally, the specimens were dried in a vacuum, and sprayed with gold using an EMS Quorum (EMS 150R ES) ion-sputtering instrument and observed through an Analytical Scanning Electron Microscope (SEM) (JEOL JSM-6010LA, Tokyo, Japan) installed with IntouchScope software (JSM-IT200 InTouchScope™ SEM Series, JEOL Solutions, Tokyo, Japan) [61].

### 4.9. Statistical Analysis

All experiments were performed on independent biological replicates. Statistical significance was determined for control and experimental groups using a paired sample t-test. Data points were excluded if contamination was identified. Statistical analyses were preformed using OriginPro Plus version 2021b (OriginLab Corporation, North Hampton, MA, USA).

## 5. Conclusions

As this facultative anaerobe continues to cause serious public health problems, the need to investigate *Salmonella* has received much scientific scrutiny [61,62,63,64]. In this study, we demonstrated the antibacterial activity of CBD against two relevant pathogenic bacteria, *S. typhimurium* and *S. newington*. Despite the scarce knowledge of the molecular mechanisms of CBD’s mode of action, we propose that the antibacterial activity might be due to membrane integrity disruption, and this was verified through the utilization of plate assays, fluorescence microscopy, and kinetic studies. These experiments confirmed that CBD has antibacterial activity against our target bacteria. Additionally, our comparative studies showed that CBD has antibacterial activity similar to ampicillin with a MIC roughly one-fifth of ampicillin. We observed the resistance development of *S. typhimurium* and *S. newington* against CBD treatment. Resistance development was observed after 48 h. These results suggest that *Salmonella* resistance to CBD might be conferred through a different mechanism than antibiotic resistance. These results posed the question of CBD–antibiotic co-therapy as a potential novel application. Finally, we observed CBD to be effective against *S. typhimurium* biofilms. This study further progresses our current knowledge on the effectiveness of CBD as an antibacterial agent and demonstrates the effectiveness of CBD against Gram-negative bacteria, *S. typhimurium* and *S. newington*.

## Figures and Tables

**Figure 1 molecules-27-02669-f001:**
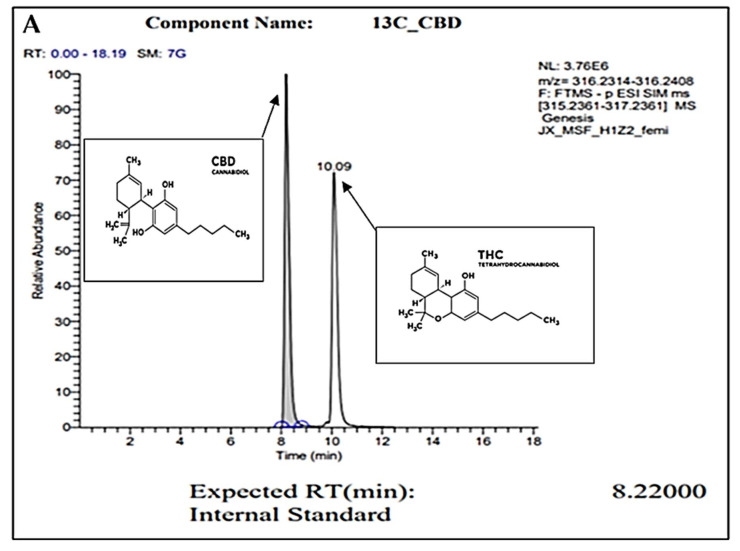
GC peaks confirm CBD as the primary compound present within our *C. sativa* extract. (**A**) represents CBD (in grey color) internal standard for GC with an 8.22000 expected RT (min). Peak detected at 10.09 min (**A**) represents the expected RT (min) of THC (in white color), which was not detected within our sample (**B**). (**B**) represents GC analysis of our CBD extract featuring an 8.2300 RT (min), confirming the sample is composed of CBD. Secondary readings around the base of the sample are consistent to those seen within our CBD internal standard (**A**).

**Figure 2 molecules-27-02669-f002:**
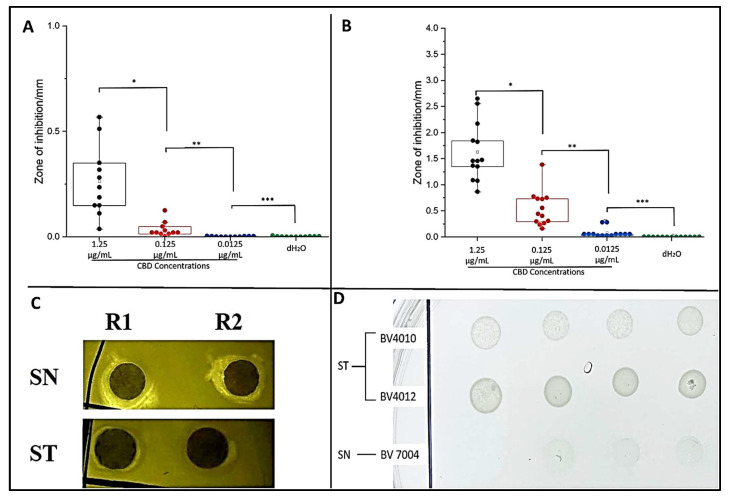
Plate assays confirm inhibitory activity of CBD. Quantitative analysis of Kirby–Bauer assay results of *S. typhimurium* (**A**) and *S. newington* (**B**). Images from Kirby–Bauer assay and CBD-produced ZOIs against *S. typhimurium* and *S. newington* (**C**). Images from spot assay featuring *S. typhimurium* and *S. newington* treated with decreasing concentrations (left to right) of CBD (**D**). (**A**) * *p*-value = 0.000961, ** *p*-value = 0.0117, *** *p*-value = 0.7179. (**B**) * *p*-value = 0.00000343, ** *p*-value = 0.000480, *** *p*-value = 0.00832.

**Figure 3 molecules-27-02669-f003:**
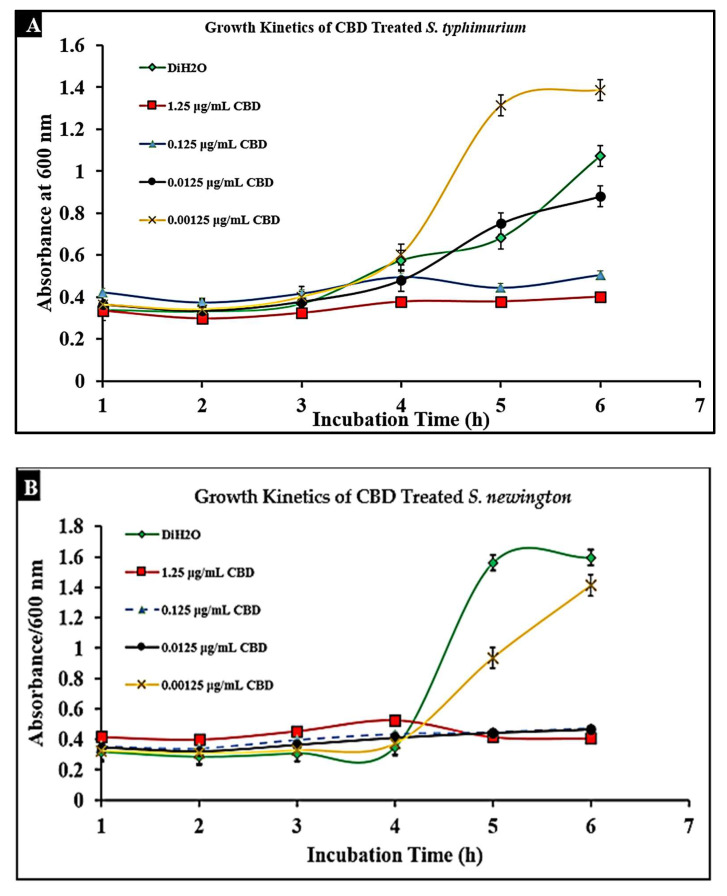
Treatment of lag-phase *S. typhimurium* (**A**) and *S. newington* (**B**) with CBD. The effect of several CBD dilutions on OD600 was recorded in triplicate.

**Figure 4 molecules-27-02669-f004:**
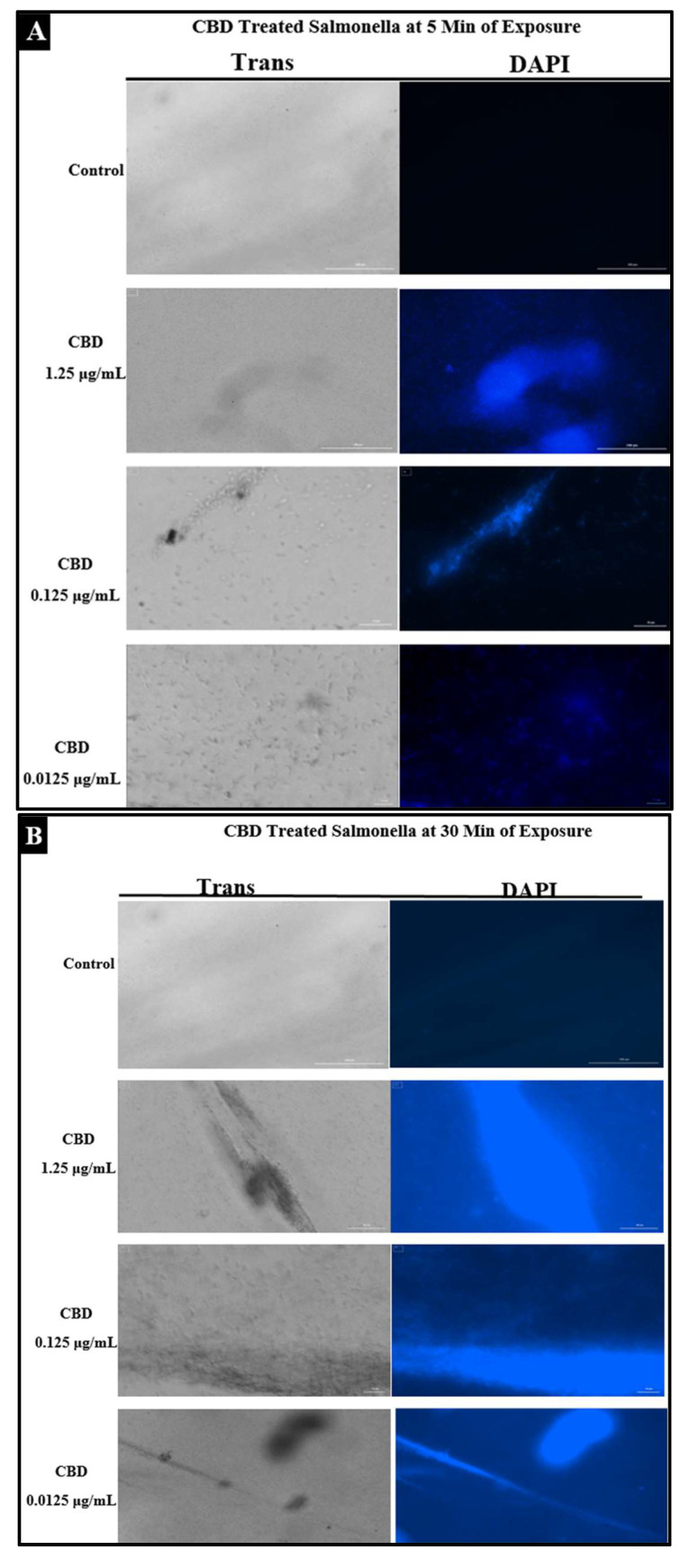
(**A**). Fluorescence microscopy of CBD-treated *Salmonella* cells stained with DAPI. Images were acquired at the time points of 5 min. DAPI staining was utilized to assess membrane integrity following treatment with CBD. (**B**). Fluorescence microscopy of CBD-treated *Salmonella* cells stained with DAPI. Images were acquired at the time points of 30 min. DAPI staining was utilized to assess membrane integrity following treatment with CBD. (**C**). Quantitative analysis of membrane integrity via densitometry analysis of fluorescence after DAPI staining.

**Figure 5 molecules-27-02669-f005:**
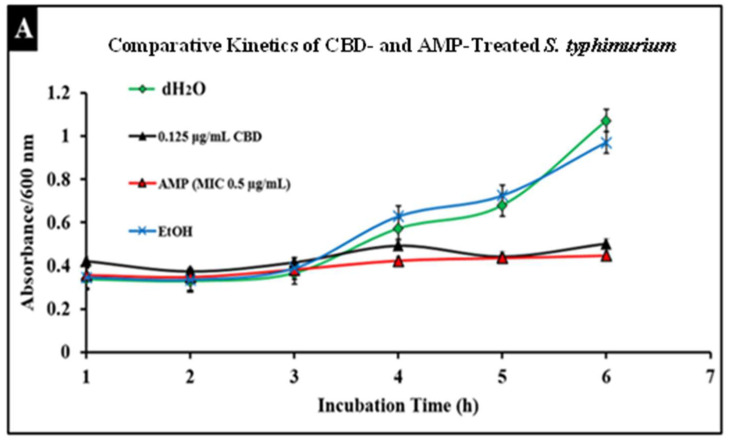
Comparative efficacy of CBD and antibiotic treatment of *Salmonella typhimurium* (**A**) and *Salmonella newington* (**B**). *Salmonella* cells were cultured overnight and then treated with either CBD or ampicillin. OD600 was measured hourly over a 6 h time period to observe bacterial growth following treatment.

**Figure 6 molecules-27-02669-f006:**
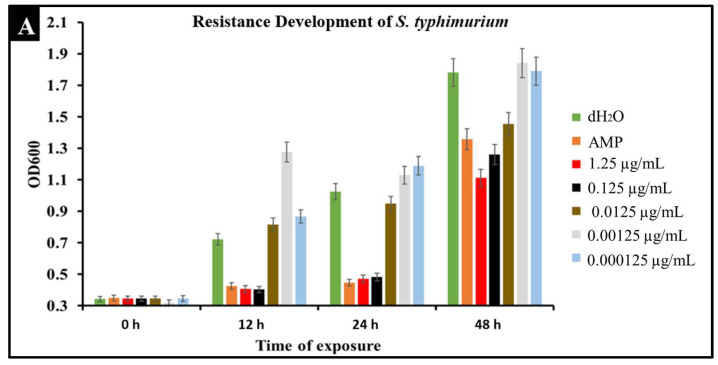
Development of resistance to CBD and ampicillin treatments by *S. typhimurium* (**A**) and *S. newington* (**B**) examined through extended kinetics. Bacterial cultures were treated with either ampicillin, CBD dilutions, or dH_2_O. OD600 was recorded over a period of 48 h.

**Figure 7 molecules-27-02669-f007:**
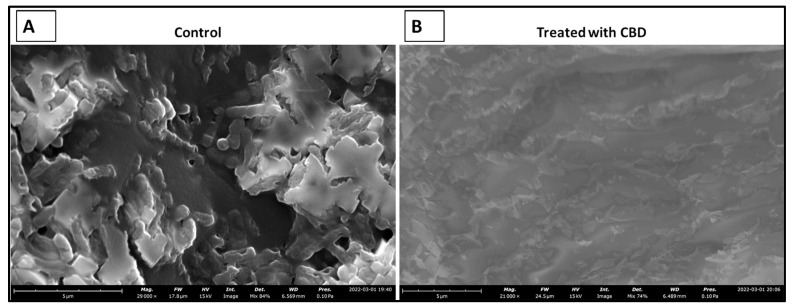
SEM images depicting the comparison between untreated (control) and treated (0.125 µg/mL CBD) *S. typhimurium* biofilms. Untreated *S*. *typhimurium* biofilm resulted in continued persistence (**A**). CBD treatment at 1.25 µg/mL resulted in a reduction of *S*. *typhimurium* characterized by fragmented cells suggesting *S*. *typhimurium* cell death (**B**).

## Data Availability

Data are contained within the article.

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
