# Peer review of "Cannabis sativa* CBD Extract Shows Promising Antibacterial Activity against *Salmonella typhimurium* and *S. newington"

_molecules, 2022, doi:10.3390/molecules27092669_

Round 1
Reviewer 1 Report
In the manuscript “Cannabis sativa CBD Extract Shows Promising Antibacterial Activity against S. typhimurium and S. Newington” the Authors examines the antibacterial potential of CBD for its application as a therapeutic agent against ampicillin resistant S. typhimurium or in a potential co-therapy designed with CBD and ampicillin. The manuscript is well written, and the results clearly shown. Given that the focus of the research falls on antibiotic-resistance topic, an emergence that has become a major concern worldwide, with considerable clinical and economic impact, there are some shortcomings on which the authors should respond:
- The study was conducted on a too small number of strains. At least in the preliminary studies it would have been appropriate to test a greater number of strains for CBD sensitivity.
- Antibiotic resistant (AR) salmonella strains were not included in the investigation. Many studies underline the antibacterial synergistic effect of natural compounds of plants origin when combined with other antimicrobials and the capability of these associations to “restore” the antibiotic activities against multidrug-resistant isolates from gram-negative species* and to decrease the concentrations of antibiotics used**.
- It is now recognized that hospital-aquired infections are caused by bacteria organized in biofilms, and the activity of any natural or synthetic compound should also be evaluated even when AR bacteria are protected inside biofilms**
- ATCC/NCTC microbial reference strains have been also not included in the study as control.
*Lorenzi V, Muselli A, Bernardini AF, Berti L, Pagès JM, Amaral L, Bolla JM. Geraniol restores antibiotic activities against multidrug-resistant isolates from gram-negative species. Antimicrob Agents Chemother. 2009 May;53(5):2209-11. doi: 10.1128/AAC.00919-08. Epub 2009 Mar 2. PMID: 19258278; PMCID: PMC2681508.
**Iseppi R, Mariani M, Condò C, Sabia C, Messi P. Essential Oils: A Natural Weapon against Antibiotic-Resistant Bacteria Responsible for Nosocomial Infections. Antibiotics (Basel). 2021 Apr 10;10(4):417. doi: 10.3390/antibiotics10040417. PMID: 33920237; PMCID: PMC8070240.
Author Response
We wish to thank reviewer to these great insights. It has helped improve the quality of our manuscript.
- The study was conducted on a too small number of strains. At least in the preliminary studies it would have been appropriate to test a greater number of strains for CBD sensitivity.
Author Response: Future studies will utilize AR strains increasing the number of strains investigated. This future research will additionally explore CBD sensitivity, CBD-antibiotic synergy, CBD delivery.
- Antibiotic resistant (AR) salmonella strains were not included in the investigation. Many studies underline the antibacterial synergistic effect of natural compounds of plants origin when combined with other antimicrobials and the capability of these associations to “restore” the antibiotic activities against multidrug-resistant isolates from gram-negative species* and to decrease the concentrations of antibiotics used**.
Author Response: Future studies will focus on antibiotic resistant Salmonella strains as this study seeks to initially confirm antibacterial activity against Salmonella in general prior to further application against resistant strains. Suggested literature has been included within the discussion discussing future studies and application.
- It is now recognized that hospital-aquired infections are caused by bacteria organized in biofilms, and the activity of any natural or synthetic compound should also be evaluated even when AR bacteria are protected inside biofilms**
Author Response: CBD effectiveness against Salmonella biofilm has now been included within the study to assess the efficacy against Salmonella in a clinically relevant form. A follow up research is currently be carried out to understand the specific mechanism orchestrating biofilm inhibition of CBD. Suggested literature has been included within references.
- ATCC/NCTC microbial reference strains have been also not included in the study as control.
Author Response: ATC/NCTC References for these strains are not available to us, these two strains are a kind gift to our Laboratory from distinguished Professors Anthony R. Casjen, and Poteete. We have duly acknowledged this information in the methods section.
*Lorenzi V, Muselli A, Bernardini AF, Berti L, Pagès JM, Amaral L, Bolla JM. Geraniol restores antibiotic activities against multidrug-resistant isolates from gram-negative species. Antimicrob Agents Chemother. 2009 May;53(5):2209-11. doi: 10.1128/AAC.00919-08. Epub 2009 Mar 2. PMID: 19258278; PMCID: PMC2681508.
**Iseppi R, Mariani M, Condò C, Sabia C, Messi P. Essential Oils: A Natural Weapon against Antibiotic-Resistant Bacteria Responsible for Nosocomial Infections. Antibiotics (Basel). 2021 Apr 10;10(4):417. doi: 10.3390/antibiotics10040417. PMID: 33920237; PMCID: PMC8070240.
These references have helped improve our knowledge of natural compounds against antibiotic resistant bacteria. We have included these references in the body of our work. Our follow up paper which is on the pipeline also will future these two articles because they have helped shape our thought and hypothesis.
Reviewer 2 Report
The work present an interesting property of a cannabidiol extract. In this case, it is a antibacterial effect. Nevertheless, there is one comment relating to the purity of the extract. It seems, looking at Figure 1 (left site) that there are more than one compounds co-eluted. The authors should use MS/MS chromatography to prove the purity of the prepared extract.
Author Response
We wish to thank reviewer for the kind words and for the pointer concerning figure 1. We appreciate your effort for going through our manuscript out of your busy schedule.
Author Response: Figure adjusted for clarity and to properly convey meaning of the results, secondary peaks consistent with those found within the internal standard for CBD shown in fig.1A.
Reviewer 3 Report
The paper is very interesting, well written and well organized. In addition, it explores a very relevant subject. The ways and means are well described as well as the obtained results which are statistically analyzed. However, in order to make it a stronger paper, I suggest some points before it can be published:
- Figure 1: legends need to be improved to be self-explanatory. And please separate it in figure 1a and 1b
- a list of abbreviations would improve manuscript
- please standardize font sizes in figures 3a and 3b, and improve figure legend.
- in figure 3a it is observed a higher growth in the presence of 0.00125ug/mL of CBD than in the medium without CBD. How authors explain that?
- According to authors, “Previous literature has outlined the potential of CBD against several, mostly gram-positive bacterial pathogens including Staphylococcus aureus, Streptococcus pneumoniae, and Clostridiosis difficile.” It would improve discussion section if authors provide some results from these works, such as MIC and relevant information to be compared with the present work
- The methodology of CBD extraction is all from reference 50? It is not clear in the text. If it is not, please provide reference for this protocol.
- Please provide references for items 4.5; 4.6 and 4.7
Author Response
We wish to thank reviewer for the kind comments, it has inspired our group, and the moral has been high since receiving these reviews. We also appreciate your great effort for going through our lengthy manuscript, out of your busy schedules.
- Figure 1: legends need to be improved to be self-explanatory. And please separate it in figure 1a and 1b
Author Response: Legends separated into A and B. Figures altered to be more self-explanatory.
- a list of abbreviations would improve manuscript.
Author Response: List of abbreviations now included in the work.
- please standardize font sizes in figures 3a and 3b and improve figure legend.
Author Response: Figures revised, and fonts standardized in figures 3A and 3B.
- in figure 3a it is observed a higher growth in the presence of 0.00125ug/mL of CBD than in the medium without CBD. How authors explain that?
Author Response: Higher growth rate at low concentration CBD treatment is now discussed within the work and compared to similar phenomena in literature.
- According to authors, “Previous literature has outlined the potential of CBD against several, mostly gram-positive bacterial pathogens including Staphylococcus aureus, Streptococcus pneumoniae, and Clostridiosis difficile.” It would improve discussion section if authors provide some results from these works, such as MIC and relevant information to be compared with the present work
Author Response: Discussion section now references MICs and other relevant information pertaining to literature referenced. Additionally compares the findings exhibited throughout our study.
- The methodology of CBD extraction is all from reference 50? It is not clear in the text. If it is not, please provide reference for this protocol.
Author Response: Referencing corrected to include relevant methodology literature for CBD extraction.
- Please provide references for items 4.5; 4.6 and 4.7
Author Response: References now provide for sections 4.5, 4.6, and 4.7.